# High-Frequency Focused Ultrasound on Quality Traits of Bovine *Triceps brachii* Muscle

**DOI:** 10.3390/foods10092074

**Published:** 2021-09-02

**Authors:** Reyes Omaro Caraveo-Suarez, Ivan Adrian Garcia-Galicia, Eduardo Santellano-Estrada, Luis Manuel Carrillo-Lopez, Mariana Huerta-Jimenez, Einar Vargas-Bello-Pérez, Alma Delia Alarcon-Rojo

**Affiliations:** 1Facultad de Zootecnia y Ecologia, Universidad Autonoma de Chihuahua, Chihuahua 31000, Mexico; p225247@uach.mx (R.O.C.-S.); igarciag@uach.mx (I.A.G.-G.); esantellano@uach.mx (E.S.-E.); lmcarrillo@uach.mx (L.M.C.-L.); mhuertaj@uach.mx (M.H.-J.); evargasb@sund.ku.dk (E.V.-B.-P.); 2Consejo Nacional de Ciencia y Tecnologia, Alcaldía Benito Juárez, Ciudad de Mexico 03940, Mexico; 3Department of Veterinary and Animal Sciences, Faculty of Health and Medical Sciences, University of Copenhagen, DK-1165 Copenhagen, Denmark

**Keywords:** high-frequency ultrasound, focused-ultrasound, beef tenderness, meat color

## Abstract

This aim of this study was to evaluate the effect of high-frequency focused ultrasound (HFFU) on quality traits of bovine *Triceps brachii*. Four treatments (0, 10, 20, and 30 min) of HFFU (2 MHz and 1.5 W/cm^2^) were applied to bovine *T. brachii* muscle. Immediately after treatment, evaluations of color, pH, drip loss, water holding capacity, and shear force in meat were undertaken. The application of HFFU slightly decreased (*p* < 0.05) the redness of meat. In addition, a significant (*p* < 0.05) decrease in the shear force of meat was observed after the application of HFFU at 30 min. No effect (*p* > 0.05) was observed on other color parameters, drip loss, and water holding capacity of meat. Overall, HFFU improved beef tenderness without negative impacts on color, pH, drip loss, and water holding capacity of meat. HFFU offers the option of tenderizing specific muscles or anatomical regions of the beef carcass. These findings provide new insights into the potential application of ultrasound in meat processing.

## 1. Introduction

Ultrasound is considered an emerging method with significant potential to control, improve, and accelerate processes without damaging the quality of food products [1]. Ultrasound refers to the sound waves beyond the audible frequency range that are generally above 20 kHz. When ultrasound passes through a liquid medium, the interaction between the ultrasonic waves, the liquid, and the dissolved gas leads to a phenomenon known as cavitation [2].

During the past decade, the use of power ultrasound has become an alternative non-thermal food processing technique. Ultrasound, alone or in combination with other methods, has been shown to be a potential tool to improve meat quality parameters, such as tenderness, functional properties of proteins, shelf life, and mass transfer [3]. Additionally, ultrasound helps to reduce the use of salt in processed meats, improve cooking, and inactivate microorganisms in meat and derived products [4].

Ultrasound has also been shown to have a positive effect on meat tenderness because it causes alterations of muscle integrity and collagen structure [5]. It can also reduce brine time without affecting meat quality, while improving salt diffusion rates [4]. It has been reported that ultrasound increases water retention capacity and cohesion in meat and meat products [5].

Ultrasound at high frequency (1–10 MHz; HFU) is used as a method to monitor the composition and physicochemical properties of food components and products during processing [6,7]. Within the frequency range of 400–600 kHz, the production of hydroxyl radicals causes oxidative degradation in an aqueous system. It is possible that these physical and chemical effects cause modifications in the tenderness and other physical properties, such as color and water retention, of meat that is subjected to an ultrasound treatment [8].

Acoustic cavitation, defined as the formation of bubbles in a medium exposed to an ultrasonic field, plays an important role in the mechanical, chemical, and optical effects of ultrasound on biological tissues [9]. These bubbles oscillate, gradually expand, and finally disintegrate by a violent and asymmetric collapse [10].

Within the field of food science, ultrasound has focused on equipment with high intensities and low frequencies (~20 kHz); however, this condition lacks the focusing characteristic when transmitting the ultrasound waves to the diffusion medium. In recent years, significant interest has been paid to therapeutic applications of ultrasound due to its ability to noninvasively penetrate deep into tissue [10].

The development of ultrasound transducer technology has led to the development of focused ultrasound technology (FUS). FUS at high intensities, known as high-intensity focused ultrasound (HIFU), has been used to mechanically fractionate soft tissues [11]. The mechanism of action of FUS is to focus the ultrasonic waves and transfer enough acoustic energy in a small area. A high degree of focus allows ultrasonic energy to pass through the tissue without causing excessive heating, leading to faster treatment times [12].

Although FUS has been mainly used in the medical area, this ultrasound technology has significant potential for application in the meat industry. Until now, no study has been reported on the use of high frequency for meat quality improvement.

Some beef muscles are tough and there is a need for alternative technologies to improve this trait. In the present study, bovine *Triceps brachii* was chosen due to its high toughness and low economic value; thus, any improvement in its quality traits would be considered an advantage. We hypothesized that the physical effects generated during high-frequency cavitation are not strong enough to produce the shear force necessary to induce changes in meat structure and composition. Therefore, the objective of this study was to determine the effect of high-frequency focused ultrasound on meat quality traits of bovine *Triceps brachii*.

## 2. Materials and Methods

Meat samples were obtained from bovine *Triceps brachii* muscle from the *Raramuri* Criollo breed slaughtered according to the Official Mexican Regulations for slaughtering domestic and wild animals (NOM-033-SAG/ZOO-2014). After 48 h post mortem, *T. brachii* was separated from the carcass, cut into 2.5 cm slices and assigned for study treatments. A Chattanooga brand Intelect^®^ NEO (DJO Ltd., Guildford, UK) ultrasound module was used for the high-frequency focused ultrasound (HFFU) application. The module was equipped with a large head (θ = 5 cm) at a frequency of 1 MHz and an intensity of 1.5 W/cm^2^ for 0 (control, non-sonicated), 10, 20, and 30 min (based on results of previous experiments), leaving the head fixed on a metal base for all the treatment times. 

Color was measured using a Minolta colorimeter (Model CR-400, Konica Minolta Sensing, Inc., Osaka, Japan) based on the color co-ordinates, namely L*, a*, b*, C*, and Hº. The Delta E (ΔE) value was calculated to obtain the difference between the color of the control and sonicated samples. Between each sonication time, temperature control was carried out by allowing the ultrasound head to cool in a container with ice and water. The area for color measurement was the entire tissue surface that had contact with the head of the ultrasound (both sides of the slice). The meat pH was recorded randomly at three different positions within the sample using a pH meter (Hanna Instruments 99163, Nusfalau, Rumania).

Meat drip loss was evaluated following the method of Honikel and Hamm [13], in which 3 g of meat samples was suspended inside a plastic container and stored for 48 h at 4 °C. Drip loss was calculated by weight difference and expressed as percentage. WHC was determined by the press method [14], as modified by Tsai and Ockerman [15]. The shear force (SF) was measured using the method described by the American Meat Science Association [16]. Briefly, samples were cooked on grills until the meat reached 70 ± 1 °C at the geometric center. Subsequently, the samples were stored at 4 °C for 24 h, then 6 cylinders with a diameter of 12.7 mm were obtained parallel to the longitudinal orientation of the muscle fibers. The cylinders were cut using a Warner Bratzler blade in a “V” shape (60° triangular opening) at a speed of 2.0 mm/s. The maximum force (Newtons) to cut each cylinder transversely was recorded with a TA-TX-plus texture analyzer (Stable Micro Systems Ltd., Surrey, UK).

Data were analyzed with the statistical package SAS 9.4 [17] using the GLM general linear model procedure (*p* ≤ 0.05). The time of ultrasound treatment was classified as a class variable. When ultrasound, time, or ultrasound × time interactions were significant (*p* ≤ 0.05), a Tukey test was performed to compare means. The results are expressed as the mean ± standard error (S.E.) of three replicates. 

## 3. Results and Discussion

A significant effect (*p* < 0.05) of HFFU treatment was observed in the color variables a* and ΔE (Table 1), and in pH and SF (Figure 1). The color variables L*, b*, C*, and Hue angle (H*) were similar (*p* > 0.05) between HFFU application times. The control (0 min ultrasound treatment) had the highest (*p* < 0.05) redness (a* = 22.05), whereas at 20 min it had the lowest value (a* = 16.70). The a* value is related to the amount of myoglobin present in the meat [18]. In this study, the ultrasound frequency and application time decreased the a* value as the sonication time of the samples increased. Although the a* values for the 10, 20, and 30 min sonication were nominally different (Table 1), they were statistically similar (*p* < 0.05) and only different from the control (0 min ultrasound) treatment.

Meat color is sensitive to ultrasound. Several reports have shown that the redness of meat is affected by different ultrasound intensities and frequencies [19,20,21]. The observed changes in color can be caused by cavitation, which may accelerate chemical, physical, and enzymatic changes in meat [22]. As a consequence, the degradation of the compounds responsible for color during ultrasound treatment may be related to oxidation reactions promoted by interaction with free radicals formed during sonication [23]. These reactions could be responsible of the changes in red color observed in meat after HFFU application at the three sonicated times (10, 20, or 30 min). Many studies have shown that ultrasound treatment does not affect color variables. In consequence, in this study, there were no significant differences between sonicated samples and control samples. It has been proposed that although the temperature increases during ultrasound treatment when using ultrasonic baths, the heat generated is insufficient for thermal denaturation and oxidation of color pigments such as myoglobin and metmyoglobin. This is because the generated heat dissipates in the diffusion medium and, thus, it is not possible to observe an effect on color variables [3].

However, in the present study, the frequency and time of sonication affected the proportions of a* in the meat. This is probably because of chemical changes in the tissue when exposed to long periods of the probe. It is also known that any increase in molecular vibration in tissue may result in the generation of heat.

The 10 min ultrasound application was the only treatment that showed a significant effect on meat pH (*p* < 0.05) (Figure 1). Ten minute ultrasonication presented the lowest pH (5.04), compared to the other treatments (5.39, 5.46, and 5.51 for control, 20, and 30 min, respectively). pH changes may be due to a release of ions from the cell structure to the cytosol [19]. Additionally, changes in the structure of proteins and the position of some ionic groups by cavitation causes the rupture of the cell membranes and the release of material from inside the cells to the extracellular medium [24]. This may explain the variation of the pH in the 10 min treatment, because this release occurs in the first minutes and no further changes in pH were found in higher ultrasound application times. This probably explains the difference between times of ultrasonication, because the intracellular compounds released from the extracellular medium could stabilize the pH at 20 and 30 min. 

No significant differences (*p* > 0.05) were found in WHC and DL of meat among treatments. The change in pH did not impact WHC and DL, probably because a higher number of samples was needed for the HFFU application. This was the first time HFFU equipment was used in meat and the technique is not yet well established. For SF, significant differences (*p* < 0.05) were observed among treatments. The control treatment showed the highest value (47.53 N), with a tendency to decrease as ultrasound time increased. The 30 min treatment presented the lowest value (33.53 N). It was found that the highest WHC value presented lower SF values. Beef tenderness is a complex trait that depends on muscle work and the presence of connective tissue. Some muscles have a large amount of collagen and a high level of crosslinking and connective tissue [25]. *Triceps brachii* is a tough muscle [26,27] but, like other muscles subjected to ultrasound application, such as *Longissimus dorsi*, *lumborum*, and *Semitendinosus* [27,28], *triceps brachii* may present a tendency to decrease its shear force. However, Got et al. [24] reported no effect of ultrasound treatment at a frequency of 2.6 MHz in the beef *Semimembranosus* muscle. This is probably because the use of frequencies above 1 MHz shows a relatively lower cavitation activity due to the temperature reached by the bubbles and an insufficient time for the cavitation bubbles to collapse [29]. Similarly, Sikes [30] found no significant differences in the shear force of the *Sternomandibularis* muscle subjected to ultrasound treatment at a frequency of 2 MHz for 10 min. The lack of effect may be attributed to the conditions used in the experiment. It is necessary to focus the energy on the target area and not dissipate it in the medium to obtain cavitation. Such a condition can be achieved using focused transducers. In the present study, the focused transducer was capable of tenderizing meat only in the sonicated area, which could be of practical importance, particularly when only specific meat areas need to be tenderized. 

The type of muscle may influence the shear force response to ultrasonication [27,30]. *T. Brachii* has a large amount of connective tissue, which does not allow the penetration of ultrasound waves. Hoogland [31] studied the penetration of high-frequency ultrasound waves on different tissues and reported that the frequency of 1 MHz has penetration values of 9 mm in muscle, 50 mm in fat, and 6.2 mm in tendon, presenting a tendency to decrease as the frequency is increased. At frequencies of 3 MHz, the wave penetration values were 3, 16.5, and 2 mm for muscle, fat, and connective tissue, respectively. Got [24] and Sikes [25] used 650 KHz, 850 KHz, and 2.6 MHz. These frequencies were too high to achieve enough ultrasound penetration and, in those studies, they did not use focused technology. The main challenge of this technique is to maximize the accumulation of energy in the targeted area, in order to induce a higher effect without causing damage to the tissue surrounding the area. The focused technology of HFFU has adopted strategies to solve this problem. One is to use generated high-energy ultrasound waves and focus them on a small point (head) [31].

Ultrasonic focused application consists of the ultrasonic transducer being brought into direct contact with the surface. However, there is a small air gap between the transducer and the receiver. This is a critical problem for transmitting the acoustic signal due to the high transmission losses of the signal to the environment; in contrast, by ensuring a continuous contact, the ultrasound waves are transmitted efficiently [32]. This problem can be solved by using a coupling medium between the transducer and the treatment surface [33]. In the case of meat, the muscle exudate may be a good coupling medium. It is not always necessary to use a coupler if the transducer is in direct contact with the food sample [34].

The thermal effect of HIFU is caused by the absorption of ultrasonic waves, which cause vibration or rotation of molecules in the tissue, and this movement results in frictional heat. Depending on the temperature and duration of contact, the tissue may become more susceptible to increased heat or to the development of protein denaturation [11,12,35]. The mechanical and thermal effects caused by acoustic cavitation may be undesirable [10], particularly in the case of meat, because high temperatures can affect the appearance of the meat, leading to consumer rejection.

Finally, the lack of differences in meat WHC and DL due to HFFU may be due to the specific conditions of this study. It is important to note that WHC and DL values were within normal values for beef [25], so no detrimental effects on meat quality were detected.

## 4. Conclusions

Ultrasound is considered to be an emerging technology that, in some conditions, improves the physicochemical quality of meat. This study determined the effect of high-frequency focused ultrasound on meat traits of a tough and low-value bovine muscle such as *Triceps brachii*. Overall, the use of HFFU at 1 MHz on *T. brachii* decreased the shear force and this effect was proportional to sonication time. The use of HFFU for 30 min yielded tender meat without detrimental changes in other technological properties. HFFU may be considered as a strategy to positively influence the tenderness of *T. brachii*. However, HFFU on different tough muscles needs to be further investigated and consumer acceptance should be included in those studies. HFFU technology allows ultrasound waves to be concentrated in a specific anatomical point in order to improve quality traits and improve the value of beef carcasses. In addition, focused ultrasound has the potential to be used in combination with other technologies, such as enzyme addition and tumbling, which are commonly used in the food industry.

## Figures and Tables

**Figure 1 foods-10-02074-f001:**
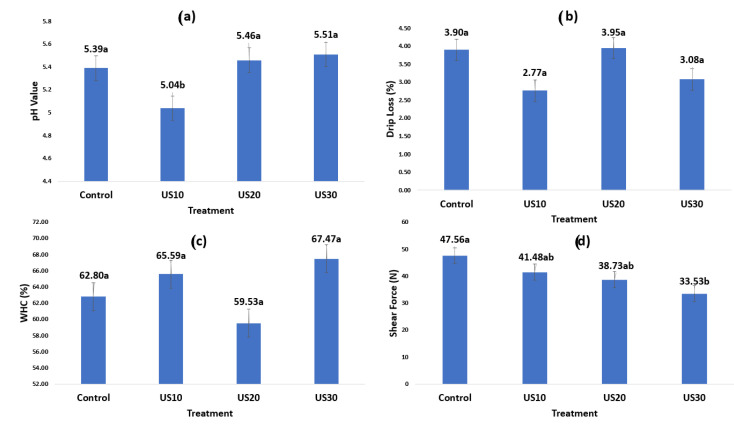
Meat quality traits after high-frequency focused ultrasound application. (**a**) pH value, (**b**) drip loss, (**c**) water holding capacity (WHC), (**d**) shear force. Treatments: Control = meat without ultrasound treatment, US10 = meat with the application of 10 min ultrasound, US20 = meat with the application of 20 min ultrasound, US30 = meat with the application of 30 min ultrasound. ^a,b^ Different letters within the same trait denote significant difference (*p* < 0.05).

**Table 1 foods-10-02074-t001:** Tukey HSD (honest significant difference) test for trichromatic variables of bovine *Triceps brachii* after ultrasound treatment at different times.

CIE L*a*b* ^1^						
Treatment ^2^	L*	a*	b*	C*	H*	ΔE
Control	36.34 ± 8.98 ^a^	22.05 ± 1.28 ^a^	12.64 ± 1.75 ^a^	25.42 ± 1.10 ^a^	29.85 ± 14.38 ^a^	0.01 ^a^
US10	38.05 ± 8.98 ^a^	17.41 ± 1.28 ^b^	10.76 ± 1.75 ^a^	20.49 ± 1.10 ^a^	31.64 ± 14.38 ^a^	6.77 ^b^
US20	37.38 ± 8.98 ^a^	16.70 ± 1.28 ^b^	11.27 ± 1.75 ^a^	20.20 ± 1.10 ^a^	33.88 ± 14.38 ^a^	7.60 ^c^
US30	37.97 ± 8.98 ^a^	19.37 ± 1.28 ^a,b^	11.65 ± 1.75 ^a^	22.64 ± 1.10 ^a^	31.15 ± 14.38 ^a^	5.26 ^d^

^1^ L* = luminosity, a* = redness, b* = yellowness, C* = chroma, H* = hue angle, ΔE = Delta-E, the color difference. ^2^ Control = meat without ultrasound treatment, US10 = meat with the application of 10 min ultrasound, US20 = meat with the application of 20 min ultrasound, US30 = meat with the application of 30 min ultrasound. ^a–d^ Different letters within the same column denote significant difference (*p* < 0.05).

## Data Availability

The datasets generated for this study are available on request to the corresponding author.

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
