# Peer review of "High-Frequency Focused Ultrasound on Quality Traits of Bovine Triceps brachii Muscle"

_foods, 2021, doi:10.3390/foods10092074_

Round 1

Reviewer 1 Report

Ultrasound has verified effect on food characteristics. Focused ultrasound technology is a promising technology in meat industry, but detailed analysis of its effects on the product (physicochemical, chemical and sensory parameters, for instance) and the efficiency and economy is needed. Therefore, the topic of manuscript can be considered as interesting for the readers. Research motivations are well defined. Introduction section is superficial. Materials and methods are given clearly, but in some cases not in details. The manuscript contains interesting and valuable results, but it needed a significant revision.

Comments:

Title of manuscript suggest a study on ’complex’ characteristics of meat, but MS has a special focus on WHC, SF and colour coordinates. It is little confusing.

Effects of ultrasound on food systems (in general), especially on meat are not discussed in Introduction section. I suggest the authors to discuss the relevant reference works.

Please give the details of colour measurement in Materials and methods section (temperature, how was the temperature controlled, how was enough ’homogenous’ surface selected etc).

Please add DOI for references (if available)

Please check the typos in the manuscript (’ 1.5 W/cm2’ in line 65, for instance).

The representation of experimental data without figures made the visibility very poor. I suggest the authors to reconsider the presentation of data.

Conclusion section is too superficial and not reflect on the main ’essence’ of the study.

Author Response

Response to Reviewer Comments:

  1. Title of manuscript suggest a study on ’complex’ characteristics of meat, but MS has a special focus on WHC, SF and colour coordinates. It is little confusing.

The title of the manuscript has been changed to: “High Frequency Focused Ultrasound on Quality Traits of bovine Triceps brachii muscle.”

  1. Effects of ultrasound on food systems (in general), especially on meat are not discussed in Introduction section. I suggest the authors to discuss the relevant reference works. ).

The effect of ultrasound on food and meat has been added and relevant studies references have been mentioned in the Introduction section.

  1. Please give the details of colour measurement in Materials and methods section (temperature, how was the temperature controlled, how was enough ’homogenous’ surface selected etc).
  2. More detalis have been added on the colour measurements in the Materials and Methods section.
  3. Please add DOI for references (if available).

DOI have been added in all references included in the References section.

  1. Please check the typos in the manuscript (’ 1.5 W/cm2’ in line 65, for instance).

I am sorry for the mistake. The superscripts have now been correctly typed.

  1. The representation of experimental data without figures made the visibility very poor. I suggest the authors to reconsider the presentation of data.

Thank you for the suggestion. Table 2 has been change by Figure 1.

8. Conclusion section is too superficial and not reflect on the main ’essence’ of the study.

Conclusions have been extended based on the main results obtained majing emphasis on the advantages and potential of the HFFU technique for the meat science sector.

Thank you.

Yours sincerely,

The corresponding authors.

Reviewer 2 Report

The presented manuscript is focused on high frequency ultrasound effects on selected quality properties of meat. The topic is interesting and novel in the area of meat science. 

The title, as well as objective, are too general and should be more precised reflecting the content of the manuscript. What was the real aim of the study? To initially check the effect of HFFU on basic meat quality traits? but why? what was the reason? if tenderization, then it should be explained clearly in the text. 

Introductory part should be more focused on HFFU theory and impact on biological tissues, especially muscle tissue.

Why bovine Triceps brachii muscle was selected for the study? At what stage of post mortem changes the muscle was excited from the carcass? What was the base for selecting these specific time of meat exposure on HFFU? 

It should be beneficial to perform sensory analysis of meat after the treatment to see if there is any impact on culinary attractiveness of the samples, as well as determine the losses during thermal treatment. 

Table 2 shows physical traits of meat not physicochemical. 

The explanation of the changes in meat color after HFFU application is not supported by the results. this needs a bit more in depth clarification and also connection with muscles and blood pigments.

Why there were no differences in WHC and DL, while pH was influenced by 10 min exposure of HFFU?

The work is interesting and has potential, maybe it needs to be revised as a short communication. 

Author Response

Response to Reviewer 2

Suggestions and Comments for Authors

  1. The title, as well as objective, are too general and should be more precised reflecting the content of the manuscript.

The title of the manuscript has been changed to: “High Frequency Focused Ultrasound on Quality Traits of bovine Triceps brachii muscle.” and it is now more precise and reflects the content of the study.

What was the real aim of the study?

As the study was done using a low value beef tough muscle such as Triceps brachii, The aim has been changed to: “the objective of this study was to determine the effect of high-frequency focused ultrasound on meat quality traits of bovine Triceps brachii.”

To initially check the effect of HFFU on basic meat quality traits? but why? what was the reason? if tenderization, then it should be explained clearly in the text. 

It is explain at the end of the Introduction section that the main beef trait we wanted to improved was ‘toughness’ therefore we choose a tough beef muscle such as Triceps brachii to carry out the study.

  1. Introductory part should be more focused on HFFU theory and impact on biological tissues, especially muscle tissue.

Sustantial information about the impact of HFFU on biological tissue has been added in the Introduction section.

  1. Why bovine Triceps brachii muscle was selected for the study?

As mention above, we have chosen T. Brachii because is a low value and tough muscle and any improvement in its quality will be important.

At what stage of post mortem changes the muscle was excited from the carcass?

Muscle was separated from the carcass a 48 h post mortem.

What was the base for selecting these specific time of meat exposure on HFFU?

Time periods of ultrasound application were based on previous experiments.

It should be beneficial to perform sensory analysis of meat after the treatment to see if there is any impact on culinary attractiveness of the samples, as well as determine the losses during thermal treatment. 

With no doubt, sensory analysis is a very important assessment and it should have been included. As this study was the first trial, we wanted to first test the effect of HFFU on the main beef traits. But, definetly, we will include sensory analysis in our next experiment. We mention this suggestion in the Conclusion section.

  1. Table 2 shows physical traits of meat not physicochemical. 

We have changed physicochemical word by ‘beef quality traits” as they are also called in meat science.

  1. The explanation of the changes in meat color after HFFU application is not supported by the results. this needs a bit more in depth clarification and also connection with muscles and blood pigments.

In discussion section, we have explained in more detail the effects of HFFU on colour making emphasis on meat pigments.

  1. Why there were no differences in WHC and DL, while pH was influenced by 10 min exposure of HFFU?

As mentioned in the discussion. We have attributed this absence of effect to the sample size used in the experiment. As this was the first time this equipment is used in meat and the HFFU technique is not well established yet.

The work is interesting and has potential, maybe it needs to be revised as a short communication. 

Thank you.

Yours sincerely

The corresponding authors.

Round 2

Reviewer 1 Report

Authors have revised the mnauscript significantly according to reviewers comments and suggestions. Amendments, rephrasings, change of title, more detialed discussion of results made the manuscript more complete and clear. i accept all answers and modifications made by the authors.